# Cognitive Analyses for Interface Design Using Dual N-Back Tasks for Mental Workload (MWL) Evaluation

**DOI:** 10.3390/ijerph20021184

**Published:** 2023-01-09

**Authors:** Nancy Ivette Arana-De las Casas, Jorge De la Riva-Rodríguez, Aide Aracely Maldonado-Macías, David Sáenz-Zamarrón

**Affiliations:** 1Graduate Studies and Research Division, Tecnológico Nacional de México/Instituto Tecnólogico de Cd. Juárez, Cd. Juárez 32500, Chih., Mexico; 2Graduate Studies and Research Division, Tecnológico Nacional de México/Instituto Tecnólogico de Cd. Cuauhtémoc, Cd. Cuauhtémoc 31500, Chih., Mexico

**Keywords:** mental workload (MWL), NASA-TLX, hierarchical task analysis (HTA), task analysis for error identification (TAFEI)

## Abstract

In the manufacturing environments of today, human–machine systems are constituted with complex and advanced technology, which demands workers’ considerable mental workload. This work aims to design and evaluate a Graphical User Interface developed to induce mental workload based on Dual N-Back tasks for further analysis of human performance. This study’s contribution lies in developing proper cognitive analyses of the graphical user interface, identifying human error when the Dual N-Back tasks are presented in an interface, and seeking better user–system interaction. Hierarchical task analysis and the Task Analysis Method for Error Identification were used for the cognitive analysis. Ten subjects participated voluntarily in the study, answering the NASA-TLX questionnaire at the end of the task. The NASA-TLX results determined the subjective participants’ mental workload proving that the subjects were induced to different levels of mental workload (Low, Medium, and High) based on the ANOVA statistical results using the mean scores obtained and cognitive analysis identified redesign opportunities for graphical user interface improvement.

## 1. Introduction

Today, the technologically complex and advanced human–machine systems being adopted in the new manufacturing environments are placing considerable mental workload (MWL) on workers. For example, mental workload increases when the complexity of the assembly of products upsurges.

Nowadays, product variants are constantly changing due to customer requirements, so more information processing, interference control, and focused attention are demanded from workers [1,2,3]. Thus, such environments’ conditions negatively impact their health, causing them to experience mental problems rather than physical problems in recent times [4]. Therefore, mental workload consideration has become relevant due to its relation to human errors, accidents, deficient performance, work stress, and other adverse effects [5]. Accordingly, in cognitive ergonomics, it is crucial to understand the influences of mental workload on worker performance and conduct analyses and interventions to reduce its harmful effects [6].

However, this poses a few challenges, such as studying and defining relevant concepts of mental workload. For example, one of the most accepted definitions of mental workload explains it as the amount of cognitive work exerted on a task over time [7,8]. Likewise, it can be understood as the relationship between environmental and task-relevant demands and the internal supply of mental resources [1]. In accordance, some authors state that mental workload exists as a function of task demands and moderating variables and can be understood as a subjective experience and a physiological reaction, resulting in task-related behavior [9].

On the other hand, the use and design of interfaces remain important mental load aspects to be studied in these new work environments. The interface can be defined as the mode through which a user interacts with the human–machine system [10]. Most of the time, these interfaces are graphical, known as Graphical User Interfaces (GUI). They can be understood as information assistance systems that workers use to acquire and enter information related to their job activities, i.e., instructions, specifications, work orders, and real-time sensor process results. These activities require cognitive resources that may be mental workload-related [1].

Additionally, GUI design considers cognitive and ergonomic factors such as user memory, physical capacities, interaction preferences, and error tolerance; therefore, an ergonomic perspective is necessary to eliminate or reduce MWL. To achieve these objectives conducting a Cognitive Analysis (CA) in the human-system interaction design phase to minimize human error and increase operator safety [11]. The cognitive analysis also implies deep task analysis, since a well-designed interface should be easy to use and feature a screen where users can quickly identify relevant information [12]. An interface should have a cognitive directness, which involves using appropriate visual cues and meaningful icons and minimizing mental transformations data, i.e., “control + shift + del + 7” to accept a statement [13].

Concerning mental workload measurement and interfaces, the Dual N-Back tasks have proven applicable to the study of mental workload. For example, a case study introduced the N-Back as a test of continuous interactive activity allowing for mental workload measurement and subject concentration [14]. A variant with an auditive stimulus known as Dual N-Back was introduced in 2003 [15].

The Dual N-Back tasks have had several uses; for example, to create a high cognitive workload and evaluate the effects of MWL on involuntary attention [16], and also to induce low, medium, and high MWL to monitor mental workload levels in office-work scenarios [17]. On the other hand, researchers used an N-Back task while measuring EEG (Electroencephalography), skin conductance, respiration, ECG (Electrocardiogram), pupil size, and eye blinks to compare measurements for a mental workload assessment [18]. Another study quantified mental workload using an EEG, functional near-infrared spectroscopy (fNIRS), and an auditive/verbal N-back task [19]. In some other research, a variant was used to examine the role of “interruptions” on qualitative and quantitative load on vigilance [20]; also, in a study using an auditory N-Back task to assess performance and cognitive states during cognitive work [21].

These studies used Dual N-back tasks to study different forms of mental workload. However, no cognitive analysis has been performed in an interface using these tasks while searching for a better user–system interaction, which is the main objective of cognitive analysis, as mentioned in different articles [22,23,24]. This work aims to design, evaluate, and validate a graphical user interface developed to create a mental workload using Dual N-Back tasks and, thus, contribute to a novel approach for the cognitive evaluation of such an interface. The study and validation of the GUI design were conducted using the NASA-TLX methodology, Hierarchical Task Analysis (HTA), and the Task Analysis Method for Error Identification (TAFEI).

### 1.1. Cognitive Analyses (CA)

#### 1.1.1. Hierarchical Task Analysis (HTA)

The importance of task analysis in improving worker productivity, in the beginning, was studied by researchers [25]. Later, a contribution concerning a valuable idea of describing a task in terms of an operations’ hierarchy and plans was generated; the result was the development of the HTA as a precedent for most human factors approaches and methods such as human error identification, workload assessment, allocation of function, interface design among others [26].

HTA enables a detailed breakdown of the tasks, which involves the hierarchical division of overall working labor into sub-tasks, elementary lessons, and specific activities expressed in a tree diagram. This methodology has been widely used and precedes any cognitive analysis [27,28,29]. One disadvantage, however, is that it can be challenging to perform [30].

On the other hand, several studies have used hierarchical task analysis to validate their work-related tasks [28,31,32,33,34]. For example, to review the goals and related cognitive processes of the mental workload-related interface [35]. Additionally, these authors incorporated the TAFEI method to study the interactions with the subjects to identify context-specific human errors.

#### 1.1.2. Task Analysis for Error Identification (TAFEI) in Interface Design

The methodology to design error-tolerant consumer products was developed and received the name Task Analysis for Error Identification (TAFEI) [36]. This method enables the prediction of interface errors by modeling the interaction between the user and the interface analyzed. 

Furthermore, it features certain advantages since it includes proposals for error reduction; however, it also requires a degree of skill to be performed effectively. Numerous studies have validated and used the methodology [37]. For example, to validate the methodology by conducting two studies to compare TAFEI’s predictions of humans’ actual performance and found that the method had an accuracy of at least 67% in error prediction [38]. In addition, to identify and decrease human errors while operating a meat grinder to improve system productivity [28]. Additionally, it was used to identify three possible error causes when changing a PC power supply [39]. Moreover, in a study, twenty-nine possible cases of illegal transfer in a “high-pressure power post” were found; each transfer could be a potential accident in this type of task [27].

### 1.2. NASA-TLX for Mental Workload Assessment

The last methodology used was NASA-TLX, one of the most widely used subjective techniques to measure Mental Workload [40]. It offers a subjective classification model to analyze workers’ perceptions of the complexity of the task. In addition, the same authors report the advantages of this technique, such as ease of use, short application time, and low cost. However, one of its disadvantages is the mental workload correlation with worker performance. Additionally, its results can be affected by the respondents’ individuality, as well as by bias, low sensibility for task differentiation, errors, and task aversion during the analysis [41].

The NASA-TLX is divided into six subscales: mental demand, temporal demand, physical demand, performance, effort, and frustration. First, each subscale’s value must be within a range of 100 points with steps of 5 points. These classifications are then merged to obtain the task load index. The other part of the test proposes establishing an individual weighting for these subscales by allowing subjects to match them in pairs based on their perceived importance [42]. NASA-TLX and HTA have been used together in some studies in different domains, including evaluating mental workload in human system design for manufacturing operations [33,43], human performance in various tasks [40,42,44], and assessment of the subjective MWL with the NASA-TLX and N-Back tasks. Researchers in a study obtained higher mental demand in a 3-Back task with average scores for NASA in a range of (38.68 ± 3.82), compared to an oddball condition with average scores for NASA in a range of (15.07 ± 1.78); these results were verified via an ocular aberration measurement as a physiological, mental workload measurement [45]. In another study, investigators created a low, medium, and high MWL by applying three variants of the Dual N-Back task to subjects in everyday life office-work scenarios [17]. The N-Back task under three demand conditions was also used in another study to determine the effect of task demand as an incentive on neurophysiological and cardiovascular effort markers [46]. Likewise, other studies used three levels of the N-Back tasks and NASA-TLX and discovered diverse levels of difficulty that were statistically different [47,48]. As can be seen, previous studies have applied Dual N-Back for various purposes; however, it is essential to mention that none of the articles conducted a cognitive analysis of the user interfaces. Therefore, this paper contributes to developing a cognitive analysis to evaluate an interface using this type of task.

## 2. Materials and Methods

This study is a transversal, quasi-experimental, and eclectic approach. It was conducted during the first semester of 2021. Accordingly, (COVID-19) pandemic restrictions applied; therefore, the present study was conducted remotely using Information and Communication Technologies (ICTs) and hardware owned by the subjects. This section will describe the participants and materials. Additionally, the GUI design method, the procedure for cognitive interface evaluation, and the statistical data analysis are presented.

### 2.1. Limitations of the Study

The study limitations were COVID-19 pandemic restrictions, time, sample size, hardware availability, and software constraints.

As mentioned, the study was completed during the first semester of 2021, when all pandemic limitations were in place. As a consequence, interactions with students by law were only virtual. This means subjects must have the internet connections, hardware, and software necessary to do the experimental sessions, and not all students comply with this requirement. All of the above influenced the reduced study sample size to carry on the project.

### 2.2. Participants

Ten participants took part in the study. All of them were healthy university students (not suffering from any current illness or mental disorder) with normal vision. Six were female, and the average age was 21.3 years (SD = 1.01 years). Before the experiment, the subjects provided their informed consent to participate in the study. The investigation was conducted following the ethical guidelines approved by the TecNM/Technological Institute of Ciudad Juarez ethics committee, which considers the General Health law on research for health Mexican regulation, the Nuremberg code, the Universal Declaration of Human Rights in its articles 1 to 5 and the American Declaration of the Rights and Duties of Men in its articles IV and XXVII. The subjects participated willingly and could suspend their participation at any time they wished; additionally, they were not reimbursed in any way. The participants were also instructed to abstain from alcohol intake 24 h before each experimental session. Participants were engineering students with certain English notions; however, all instructions were given in Spanish.

### 2.3. Materials

#### 2.3.1. Hardware

The central computer used for the interface design was a DELL computer with an Intel(R) Core (TM) i5-8250U CPU@ 1.60 GHz 1.80 GHz, an 8.00 GB RAM installed, and a Windows 10 Pro Operative system; the subjects were also connected to the main computer via the TeamViewer^®^ software, which enables remote access and can be freely downloaded from its website for non-commercial purposes. For data analysis and connection via Google Meet^®^ with the subjects, the equipment used was an ACER computer with an Intel(R) Core (TM) i5-8250U CPU@ 1.60 GHz 1.80 GHz, an 8.00 GB RAM installed, with pencil compatibility and tactile function with ten tactile points, and a Windows 10 Home operative system. Finally, for connections with the subjects via Google Meet Version 62.0.372156507 and the NASA-TLX iOS app. Ver. 1.0.3, an Apple iPad Air (3rd generation) with IOS version 14.4.2 software, was used.

#### 2.3.2. Software

The software used by this study to do a subjective mental workload measure was the NASA TLX iOS EULA app Ver. 1.0.3, developed at the Ames Research Center [49]. The software is a full computational version of its predecessor Windows NT; a pencil and paper version of this app can be downloaded from https://itunes.apple.com/us/app/nasa-tlx/id1168110608, (accessed on 19 January 2021) and data collection can only be performed through iOS System equipment.

#### 2.3.3. Methodology for GUI Design

The interface was designed using MATLAB’s GUIDE tool, a user interface design environment used widely to develop GUIs [50,51]; the researchers chose it because it offers the advantage of having a drag and drop features and programmatically generating a sequential application that uses a timer to generate a fixed time base of 3 s. This is the usual period between mental load test events, such as showing random operands (1 to 10) so that their positions are in a 3 × 3 matrix and their values are memorized [17]. While the operands are displayed in the array (for 3 s), its corresponding position and value must be remembered.

The program will then request that the position of each operand in the 3 × 3 matrix be mentally identified and decide whether an operand in time (n) appeared in the same position as an operand in times: (n−1), (n−2), or (n−3). The program can also request that an arithmetic operation (sum, subtraction, multiplication, or division) be performed under the same pattern, that is, that one of the arithmetic operations be performed mentally with the value of the current-time operand (n) and the value of any of the operands at times (n−1), (n−2), and (n−3). The mental activity performed by the user depends on the defined difficulty settings, which, following the Dual N-Back task, can be the following [17]:Easy: It consists of identifying the position and adding and subtracting the operands in times (n) and (n−1).Medium: It consists of performing addition or subtraction operations of the operands in times (n) and (n−1) or (n−2).Hard: It consists of performing one of the following operations: addition, subtraction, multiplication, or division of the operands in times (n) and (n−2) or (n−3).

After variable initialization, the program generates operators with their position every 3 s in an n-time. However, that n-time is compared to the vectors of previously configured values depending on the degree of difficulty selected. The Graphical User Interface block program diagram is shown in Figure 1.

When the number of test operations (N) is terminated, the program evaluates the user’s performance (correct/N) and terminates.

In addition, the program makes use of GUIDE, which contains the graphical interface functions such as the MATLAB© command “uicontrol,” which can insert elements such as:text. Provides static text in the figure to provide information to the user and indicate values.popup menu. Creates a menu that displays a list of options on whether the operands at times n and (n−1), (n−2), and (n−3) are in the same position or not. The user selects them by pressing the mouse button on the desired option.edit. An editable text field is created where the keyboard cursor becomes active and blinking, waiting for information to be entered. This is where the user writes the result of the mental operation he performs (addition, subtraction, multiplication, division).axes. Creates a Cartesian axis in the figure for images to be inserted. It is used to insert the icons shown to users to tell them what to do, for example, Memorize, Add, Subtract, Multiply, and Divide. It also indicates whether an answer should be given and whether it is correct or incorrect.uitable. Creates a table component in the figure for the user interface. It is used to display the 3 × 3 matrix where the operand is positioned.sound. This command lays an audio file (.wav) on the computer speaker. Audio clips simultaneously play as icons are displayed to instruct users regarding the activity they must perform, for example, Memorize, Add, Subtract, Multiply, or Divide.

### 2.4. Procedure for Interface Cognitive Evaluation

The Graphical User Interface design cognitive evaluation and validation was descriptive, analytical, quantitative, featuring a transversal design, and was undertaken in the following five phases:

#### 2.4.1. Phase 1: Task Analysis and Hierarchical Task Analysis (HTA)

In this phase, two subjects ran the interface completely and were analyzed. The HTA was developed from this analysis following the methodology proposed by Stanton [52], which includes deciding on the aim of the analysis, establishing task objectives and performance criteria, identifying sources of task information, and obtaining data and a draft decomposition diagram.

#### 2.4.2. Phase 2: Task Analysis for Error Identification (TAFEI)

As part of the cognitive evaluation, error identification was relevant in the interface design. Therefore, the GUI was evaluated to propose pertinent changes and improvements. Accordingly, the TAFEI method was helpful since it gives a vision of the error prediction and modeling of the interaction between the user and the interface under analysis [50].

Two steps were followed in this phase, according to Prabaswary [53]:Step 1: Create space–time diagrams (SSD), including the list of states that may happen in the interface. These states are constructed to represent the interface behavior as a whole (artifact), and they consist of a series of states that the interface passes through, from a starting state to the goal state. For each series of states, there will be a current state and a set of possible exits to other states.Step 2: Create the transition matrix, including the detection of impossible transitions denoted by the (-) symbol, possible and desirable denoted by (L), and possible but undesirable/Illegal denoted by (I). These transitions will be further analyzed. TAFEI intends to assist the interface (artifact) design by illustrating when a state transition is possible but undesirable (Illegal, I). Consequently, designers will try to make all the illegal transitions impossible and facilitate the cooperative attempt of interface use. All possible states are entered as headers on the matrix, where the cells represent the state transitions.

#### 2.4.3. Phase 3: Experiment for Interface Validation of Mental Workload Using Dual N-Back Tasks

In this phase, the participants take part in six sessions (two for each Mental Workload level). Previous to the first session, an informed consent form received via e-mail had to be signed and returned before moving forward. Beginning the first session, the subjects received initial information through a PowerPoint^®^ presentation. After that, they were asked to connect to the main computer via TeamViewer^®^ to begin customized training on the interface use and the tasks performed during the test. Then, they performed the assigned tasks, designated using a randomized factorial design. The subjects were identified with an “S” and a randomly assigned number from 1 to 10; T1 was used for tasks with a low mental workload, T2 for a task with a medium, and T3 for tasks featuring a high mental workload. The final schedule is shown in Table 1.

#### 2.4.4. Phase 4: Mental Workload Subjective Evaluation Using NASA-TLX

As was mentioned before, the MWL evaluation was conducted using the NASA-TLX iOS app, version 1.0.3^®^, which was downloaded in an iPad^®^ Air 3er Gen. Due to the COVID-19 pandemic, the test was conducted remotely, and the subjects contacted the researchers via Google Meet^®^. Once the assigned task was finished, the subjects responded to NASA TLX.

Since the previous application was only available in English, the subjects received a translation of all the app’s terms and definitions in advance. Then, during the test, the researchers read out each screen shown to the subjects in Spanish. In addition, to improve the comprehension of the method, NASA-TLX dimensions were explained carefully before the subjects began with the pairwise comparisons and final dimension scores assignation.

#### 2.4.5. Phase 5: Statistical Analysis

Statistical data analysis was conducted to confirm that the three levels of mental workload induced by the N-Back tasks were appropriately differentiated by subject and by software by using an ANOVA Lineal General Model with a confidence level of 95% with two factors: mental workload level and subject. The mental workload level factor had three values (Low, Medium, and High), and the subject factor had ten values (Participants).

## 3. Results

This section will present the results related to the Graphical User Interface design, HTA, and TAFEI methodologies the mental workload subjective examination completed with NASA-TLX and its corresponding statistical analysis.

### 3.1. Results of Graphical User Interface (GUI) Design

The GUI design was performed using the MATLAB© interface GUIDE. Figure 2 shows the GUI’s initial skeleton.

The interface’s main window allowed users to access the tool’s functionalities. The interaction with the subjects sought to minimize the subjects’ errors and operational problems

Visual cues and icons in the main window were divided by their functionalities into three areas:3 × 3 MatrixResponse boxOperation

Figure 3 shows an example of the GUI task in which the subject is asked to subtract the operands displayed in times (n) and (n−1).

The designed interface at the end of the test shows the two last screens. In the first one, shown in Figure 4a, users could see their performance expressed in a percentage obtained through a correct answer ratio. In the second one, shown in Figure 4b, a graph appeared showing the performance over time (seconds); the correct answers were displayed in green bars, and a number was represented in the positive part, while the incorrect responses were displayed in red bars shown in the negative aspect of the graph. Even though knowing their performance may influence the NASA-TLX answers, this limitation was not studied sufficiently to have an accurate report about it.

### 3.2. Hierarchical Task Analysis (HTA) Results

The first method applied was the HTA. The analysis broke down the main activities into smaller tasks: five subtasks and twelve sub-sub tasks resulting from such division. Using this method, sub-task five (5.0) was assigned to the execution of the test and was, thus, recognized as the most complex task since this is where the low, medium, and high mental workloads were induced via the GUI used by the participants. Likewise, this task was also considered as the one where more errors were expected to occur, especially in the high mental workload interface tasks (5.3), which demanded a higher complexity in the tasks of memorizing and calculating. Table 2 shows an excerpt of the hierarchical task breakdown; the complete HTA is reported in chart form in Appendix A, Table A1.

### 3.3. Task Analysis for Error Identification (TAFEI)

Once the HTA was carried out and the participants’ interaction and activities in the system were clarified, human errors were identified using TAFEI. Meanwhile, the determination of the SSDs was also possible. Figure 5 shows the corresponding nine SSDs in the man–machine system during the task. These diagrams revealed potential human and technical system errors that were tended to so that they could be prevented during the experimental session, as all of them could jeopardize the test and be time-consuming for both participants and researchers.

Aside from the SSDs, the transition matrix for TAFEI shown in Table 3 shows eight illegal transitions (I). These transitions are the following: from state 2 (computer on) to 4 (computer mouse not working), this error occurs when the participant moves the mouse before checking whether the software executable is available. A similar transition is an illegal transition from 2 (computer on) to 6 (keyboard not working). About the illegal transitions from 2 to 7 (test interface) and 8 (interface not working) can become potential human and system errors. Therefore, the researchers designed the training interface for the first session. As can be seen, all the illegal transitions can happen before the training or the actual test and correspond to the HTA in subtasks 4.0 and 5.0. Therefore, when these illegal transitions were identified, the researchers eliminated them and discussed them with the subjects before the tests began. Hence, such errors were absent when the actual sessions took place.

### 3.4. Results for Mental Workload Interface Validation of Mental Workload Using NASA-TLX and Statistical Analysis

Phases 3 and 4 were necessary to validate the interface using Dual N-Back tasks to analyze mental workload using NASA-TLX. The NASA-TLX adjusted results were obtained by dimension (pairwise comparison/weight) in conjunction with the overall global NASA-TLX scores. The performance average is shown in the percentage of correct responses for all participants (Table 4). The results found for overall NASA-TLX scores were in the corresponding ranges of Low (0–29), Medium (30–49), and High (50–100), values stated according to the data collected [54] and very similar to the values expressed in other studies [53].

These results show the highest average values of NASA’s dimensions in the highest mental workload level, except for the physical demand. In this respect, the execution of Dual N-Back tasks using the GUI demands physical related to computer interaction such as computer mouse movement or the keyboard use when typing a response. Accordingly, in their responses, subjects in the sample perceived the lowest values of physical demand dimension in the MWL induced by these tasks in all levels.

As a result of the ANOVA, a significant difference (*p* = 0.001 α = 0.05) was found between the mental workload levels and between NASA-TLX global score results between subjects (*p* = 0.000 α = 0.05), the results are shown in Table 5. From these results, it can be inferred that the GUIs using the Dual N-Back tasks and methods to induce MWL at three levels.

Figure 6 and Figure 7 show the overall NASA-TLX results by session. The code used included the letter “S” for the Subject, the number assigned to them randomly (1 to 10), and the number of the type of mental workload test (1: Low, 2: Medium, and 3: High). Thus, “S1–3” was used to designate subject number one, performing the high mental workload task (3). Additionally, the NASA TLX scores for the “LOW” mental workload level bars are displayed in green, the scores for the “MEDIUM” mental workload level in blue color bars, and the scores for the “HIGH” mental workload level in aqua color bars. As can be observed, participants’ highest scores for NASA TLX correspond to the induced “HIGH” mental workload level. Similar results can be seen for the rest of the mental workload levels. Therefore, it can be inferred that the interface design can induce low, medium, and high MWL levels effectively.

## 4. Discussion

This work aimed to design and evaluate a GUI developed to create a mental workload (MWL) based on the Dual N-back tasks [17]. This objective was accomplished since there are a lack of studies that evaluate interfaces using Dual N-Back tasks from a cognitive ergonomics perspective. Therefore, this paper’s novelty lies in that it offers the cognitive evaluation of such an interface, thus increasing the knowledge of the GUI, the methodology, and the participants’ performance. Additionally, the human error method and its results helped correctly identify human and system errors, as has been completed in others researches [27,28,32,36,39]. Another objective was to validate the design of the GUI with a cognitive approach. For this purpose, the Hierarchical Task Analysis (HTA) in its chart form helped break down the tasks into subtasks of interface interaction. This method preceded the NASA-TLX analysis; which was applied to determine the participants’ MWL during the interface interaction, obtaining results similar to experimentations that used interfaces and NASA-TLX [55]. The Task Analysis Method for Error Identification (TAFEI) was also applied, identifying potential human and system errors related to the state of the hardware and software. In addition, it helped identify human and system errors that can be prevented and avoided effectively.

## 5. Conclusions

As conclusions, the interface design using the Dual N-Back tasks accomplished its purpose of inducing different levels of mental workload (Low, Medium, and High) based on the ANOVA statistical results using the mean scores obtained by global NASA-TLX scores in the sample of subjects of this study. Additionally, the subject’s performance was coherent with the level of MWL induced: it was better in the Low mental workload sessions, with an average of 89.22% of correct answers, compared to an average of 80% for the Medium level, and 76.1% for the High one. Thus, it can be concluded that the Graphical user interface (GUI) design and evaluation may be effective when studying mental workload under the stated conditions. Future studies can include a bigger sample size using this graphical user interface to obtain reliable measures of mental workload by level to support additional analyses related to it and several important performance variables of participants, such as the number of correct answers and the response time in seconds. Finally, some physiological variable monitoring can be incorporated into the methodology to study the effects of mental workload on the participants under several conditions of interest.

## Figures and Tables

**Figure 1 ijerph-20-01184-f001:**
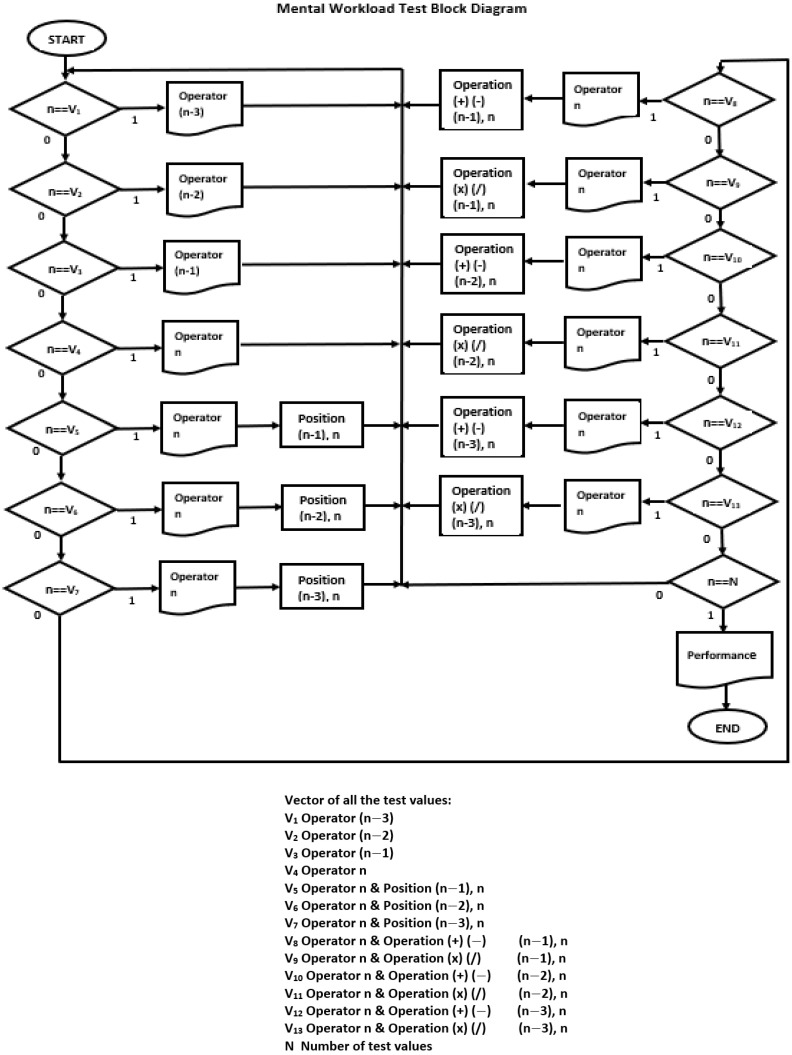
Mental Workload Program block diagram.

**Figure 2 ijerph-20-01184-f002:**
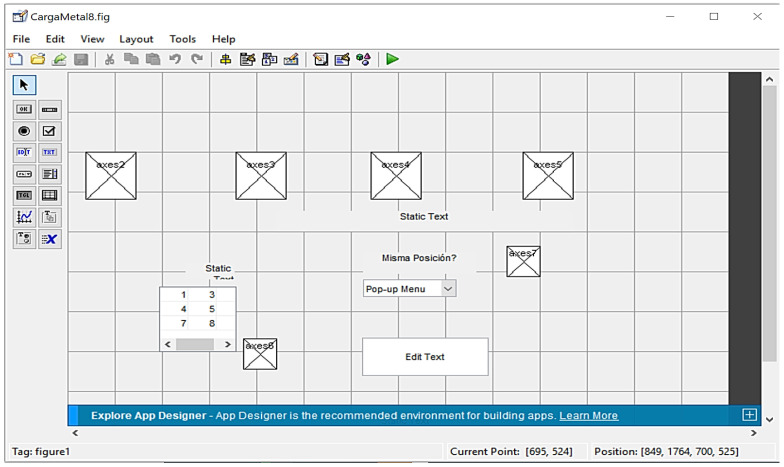
Mental Workload Graphical User Interface MATLAB© ™ interface GUIDE. Source: Authors.

**Figure 3 ijerph-20-01184-f003:**
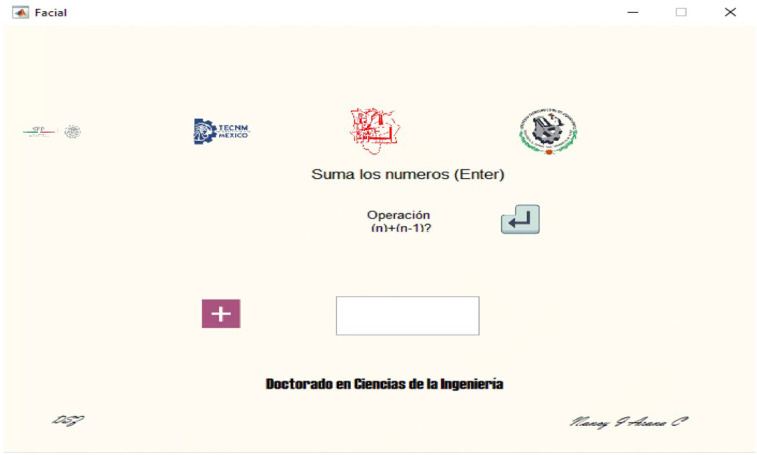
Mental Workload Graphical User Interface task window in Spanish.

**Figure 4 ijerph-20-01184-f004:**
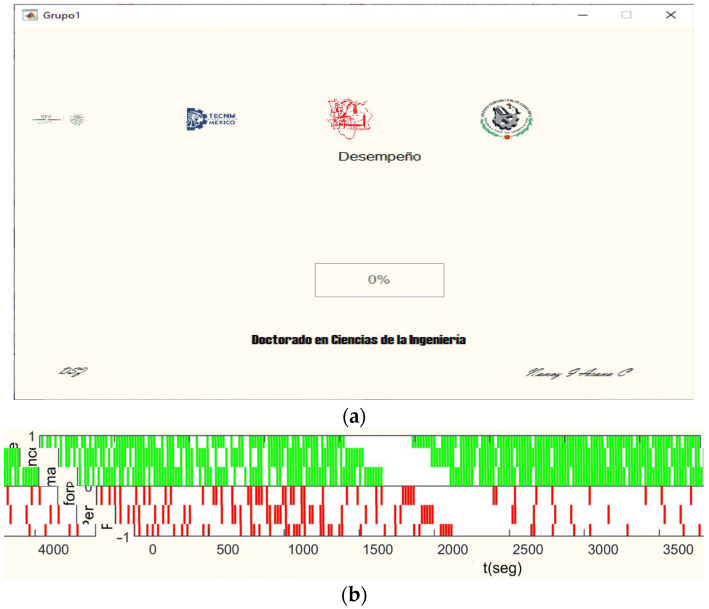
(**a**) Mental Workload GUI screen showing performance in percentage in Spanish; (**b**) Mental Workload GUI screen showing performance over the session time (rights and wrongs).

**Figure 5 ijerph-20-01184-f005:**
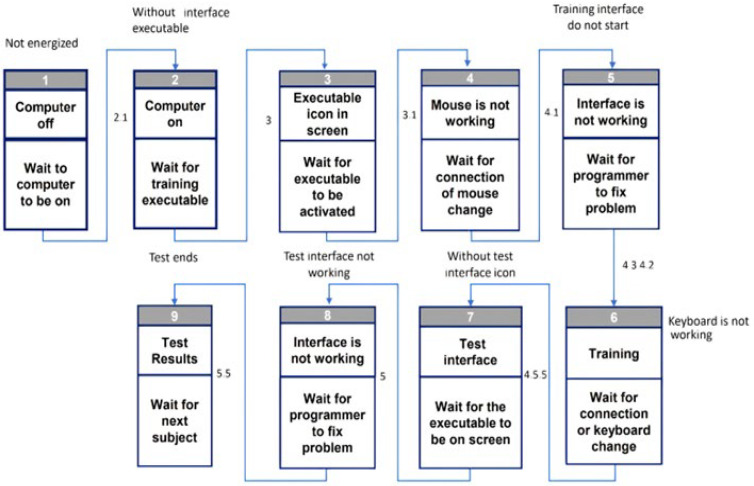
TAFEI method for human error analysis.

**Figure 6 ijerph-20-01184-f006:**
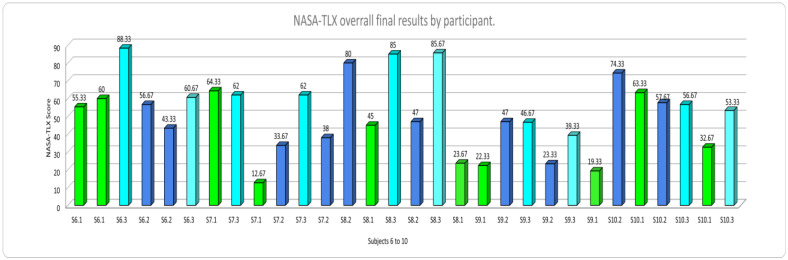
NASA-TLX overall results Subjects 1 to 5.

**Figure 7 ijerph-20-01184-f007:**
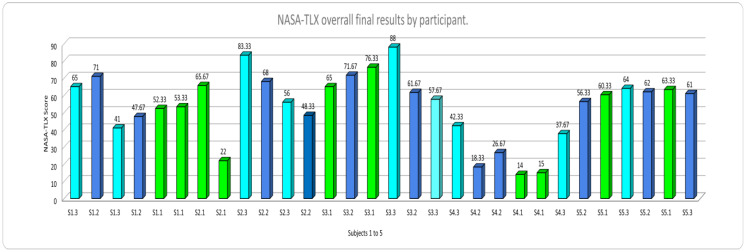
NASA-TLX overall results subjects 6 to 10.

**Table 1 ijerph-20-01184-t001:** Subjects’ session schedule.

	03/03/21	03/04/21	03/05/21	03/08/21	03/09/21	03/10/21	03/11/21	03/12/21	03/15/21	03/16/21
17:00	S6-T1	S3-T1	S4-T3	S2-T1	S10-T1	S3-T2	S8-T2	S3-T3	S2-T3	S5-T3
18:00	S10-T2	S7-T3	S1-T1	S7-T2	S7-T3	S3-T1	S2-T1	S8-T1	S1-T2	S2-T2
19:00	S7-T1	S5-T2	S7-T1	S4-T2	S7-T2	S9-T1	S5-T1	S6-T1	S6-T3	S4-T2
	03/17/21	03/18/21	03/19/21	03/22/21	03/23/21	03/24/21	03/25/21	03/26/21	04/12/21	04/13/21
17:00	S8-T3	S6-T2	S1-T3	S8-T2	S2-T3	S9-T2	S2-T2	S8-T3	S4-T3	S1-T3
18:00	S3-T2	S4-T1	S4-T1	S9-T3	S5-T2	S10-T3	S9-T3	S6-T3	S10-T3	S5-T3
19:00	S6-T2	S10-T2	S9-T2	S1-T2	S1-T1	S10-T1	S5-T1	S3-T3	S8-T1	S9-T1

**Table 2 ijerph-20-01184-t002:** Excerpt of the hierarchical breakdown into the list of sub-tasks, elementary tasks, and specific activities.

Subtask	Elementary Task	Specific Activity
4. Training Plan 4: do 4.1, 4.2, 4.3, 4.4, and 4.5 in order.	4.1 Beginning of the training	4.1.1 Click the start button
	4.2 Digit position training Plan 4.2: 4 times ((Do 4.2.1 & 4.2.2 × 3 then 4.2.3, 4.2.4 & 4.2.5 in order))	4.2.1 Listen “Memorize.”
		4.2.2 See 3 × 3 matrix with a number from 1 to 9. Memorize the digit and position
		4.2.3 Observe writing “Same Position”. (n)/(n−1)?
		4.2.4 Select “no” or “Yes” as appropriate
		4.2.5 The subject receives feedback

**Table 3 ijerph-20-01184-t003:** Transition matrix for TAFEI.

	1	2	3	4	5	6	7	8	9
1	‘	L	‘	‘	‘	‘	‘	‘	‘
2	‘	‘	L	I	‘	I	I	I	‘
3	‘	‘	‘	L	‘	‘	I	‘	‘
4	‘	‘	‘	‘	L	I	I	‘	‘
5	‘	‘	‘	‘	‘	L	I	‘	‘
6	‘	‘	‘	‘	‘	‘	L	‘	‘
7	‘	‘	‘	‘	‘	‘	‘	L	‘
8	‘	‘	‘	‘	‘	‘	‘	‘	L
9	‘	‘	‘	‘	‘	‘	‘	‘	‘

**Table 4 ijerph-20-01184-t004:** NASA-TLX Results.

	Low MWL	Medium MWL	High MWL
Mental demand	161.5	194.75	249.25
Physical demand	19.25	32.25	24
Temporal demand	214.5	176.25	235.75
Performance	57.25	105.25	172.5
Effort	99.75	113	143
Frustration level	112.25	153	169.25
Overall mental workload score (NASA TLX)	44.29	51.63	61.78
Performance Average (Correct responses)	89.22%	80%	76.1%

**Table 5 ijerph-20-01184-t005:** Mental Workload Statistical Analysis results.

Source	DG	SSR	Adj. MS	F Value	*p*-Value
MWL Level	2	3083	1541.7	7.76	0.001
Subject	9	10,306	1145.1	5.76	0.000
Error	48	9535	198.6		
Lack of adjustment	18	3285	182.5	0.88	0.608
Pure error	30	6250	208.3		
	59	22,925			

DG.: Degrees of Freedom SSR.: Adjusted sum of Squares MS Adjusted: Adjusted Mean Square.

## Data Availability

The database data used to support the findings of this study are available from the corresponding author upon request.

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
