# Peer review of "Cognitive Analyses for Interface Design Using Dual N-Back Tasks for Mental Workload (MWL) Evaluation"

_ijerph, 2023, doi:10.3390/ijerph20021184_

Round 1
Reviewer 1 Report
The article explores the cognitive analyses for interface design in determining the best approach for mental workload evaluation. Overall are good.
The followings are specific comments and some suggestions for improvement.
Abstract: Acceptable
Introduction: the statement on "the new manufacturing environments" needs to be revised because there is no extended discussion in the text. If any, add some discussion on the appropriate categories of manufacturing environments such as quality inspection activities in the precision machining process.
Materials and Methods: Acceptable, however, need to improve the numbering style. There are too many numbers that represent numbers. Better to use a unique acronym/ abbreviation. Figure 1, font size in Figure 1 is too small and unable to read in the A4 size printed version. Figure 2 needs to be improved as well as the quality of the figure. This comment is also for figure 3 and 4.
Result: Overall is good. Missing value for high in sub-topic 3.4. Re-align the source in Table 5. Improve the quality of Figure 7. Otherwise, re-draw it using Microsoft Excel or any.
Conclusion: Acceptable.
Reviewer 2 Report
Dear Authors,
the paper is promising but somehow lacking.
Some comments about the paper:
1. The quality of the figures is not so good, e.g. figure 1 is unreadable.
2. Line 128-129: one subscale is missing.
3. The number of participants in this study seems too small.
4. The paper's actual aim is unclear: is it to evaluate a GUI, or to identify errors?
Reviewer 3 Report
This paper aims to design, evaluate and validate GUI created to manipulate user’s mental workload by Dual N-Back tasks. For the study validation NASA-TLX questionnaire, Hierarchical Task Analysis (HTA), and the Task Analysis Method for Error Identification (TAFEI) were applied.
The investigation is interesting, however the sample size is very doubtful for me. The conclusions are stated on the basis of the results from 10 subjects derived from homogenous group (healthy, young students). Therefore, the paper should be completed with the analysis of research limitations, including sample size. However in my opinion the conclusions cannot be treated so strongly, and the future research are needed for confirmation the obtained results.
Moreover, if I understood well the explanation (line 280), the study was conducted in English version only, and participants received the Spanish translation before taking part in the experiment. Therefore the level of English fluency of participants could be a significant factor influencing mental workload of participants, and should be monitored and reported.
Regarding the methodical aspects of the investigation, the presentation of subject’s individual results before NASA-TLX method applying could influence participant’s opinion stated in the questionnaire. In my opinion, to avoid this effect these results should be presented at the end of the experiment. As the experiment is already done, this is one of the research limitations. Moreover, the wrote “the results show an increment in all average values of the methodology dimensions in most cases, except for the physical demand” (line 350-351), but the relation is not clear for Temporal Demand dimension.
The information about study approving by the Ethics Committee of the Tecnológico Nacional de México / I.T. 427 Cd. Juárez is not full and need to be revised.
Some editiorial remarks:
· Figure 1-4 have low resolution and therefore they are not fully readable. This is particularly important for Figure 2, as it explains the way the experiment was conducted.
· Line 231 – uitable, probably UI table
· Line 232 – sound, probably Sound
· Line 335-336 – please check the spacing and the table layout.
Reviewer 4 Report
The manuscript is written well and has a potential to be published. However, some parts are needed to be improve:
1. The abstract must be improved, results must be included.
2. The discussion section must also be improved, supporting literatures are needed.
3. Discuss why the study only has 10 participants.
Round 2
Reviewer 2 Report
Dear Authors,
the aim of the paper is now much more intelligible.
The new subsections give clearness to the reading.
The responses to the comments are exhaustive and convincing.
Reviewer 3 Report
Thank you for the answer and review comments. I am satisfied with the improved paper. However, please check the styple and typos in a new paragraph (line 412-421) and in the line 459 spacing is missing.